

# Design of oral English teaching model based on multi-modal perception of the Internet of Things and improved conventional neural networks

Haitao Qin

College of Foreign Studies, Hubei Normal University, Huangshi, Hubei, China

## ABSTRACT

Oral English instruction plays a pivotal role in educational endeavors. The emergence of online teaching in response to the epidemic has created an urgent demand for a methodology to evaluate and monitor oral English instruction. In the post-epidemic era, distance learning has become indispensable for educational pursuits. Given the distinct teaching modality and approach of oral English instruction, it is imperative to explore an intelligent scoring technique that can effectively oversee the content of English teaching. With this objective in mind, we have devised a scoring approach for oral English instruction based on multi-modal perception utilizing the Internet of Things (IoT). Initially, a trained convolutional neural network (CNN) model is employed to extract and quantify visual information and audio features from the IoT, reducing them to a fixed dimension. Subsequently, an external attention model is proposed to compute spoken English and image characteristics. Lastly, the content of English instruction is classified and graded based on the quantitative attributes of oral dialogue. Our findings illustrate that our scoring model for oral English instruction surpasses others, achieving the highest rankings and an accuracy of 88.8%, outperforming others by more than 2%.

## INTRODUCTION

In light of the pandemic, the shift towards distance education has been evident in teaching assignments. The unique teaching methodology of oral English instruction poses challenges in terms of effective monitoring over the network. Thus, there is an urgent requirement for a technique that can utilize audio and video information to assess the quality of instruction. By harnessing the full potential of IoT technology, audio and video sensors, along with other devices, can capture real-time audio and visual data from monitoring or interactive entities. This enables us to establish pervasive connections between various entities, as well as relationships between entities and individuals (*Luo & Wu, 2017*; *Wei, Wu & Cui, 2019*).

Recently, evaluation algorithms for image and sound have predominantly relied on native classification techniques. Deep learning algorithms play a crucial role in determining the final score by categorizing images or sounds. While these scoring methods

Corresponding author
Haitao Qin,
morganqin@hbnu.edu.cn

are efficient, concise, and exhibit high accuracy, they are limited in their ability to process image and sound features simultaneously, necessitating separate calculations. However, by constructing a hierarchical network framework, deep learning models can extract discriminative high-level features in a layered manner, leading to improved classification and recognition outcomes. Various deep models, including stacked autoencoders, deep belief networks, and recurrent neural networks (RNNs), have been employed in image classification research. Notably, these models demonstrate superior performance when adequate samples are available. For instance, *Zhang et al. (2022)* achieved improved results in the classification of wild bacteria through enhancements to ResNeXt50 (*Xie et al., 2017*). *Wang & Xiao (2021)* enhanced feature extraction by incorporating multiple attention mechanisms into RA-CNN. However, this model faces limitations in capturing long-distance dependencies due to the small receptive field, thereby restricting its feature extraction capabilities. Convolutional neural networks (CNNs) have emerged as mainstream models for scene processing and analysis. However, they still exhibit some deficiencies (*Wang et al., 2021*): (1) CNN operations struggle to model long-distance feature relationships; and (2) CNNs struggle to learn global semantic information. In contrast, the Transformer model can effectively process global semantic features by transforming images into sequences of image patches. The Vision Transformer (ViT) method, introduced by *Dosovitskiy et al. (2020)*, extracts global image features by leveraging the self-attention module (SA) to capture long-distance dependencies, leading to significant improvements in classification accuracy. However, ViT is limited to capturing correlations between pixels within a single image sample, resulting in suboptimal extraction of output features and generating a large number of model parameters. Moreover, the model employs class patches, which are output by the last Transformer, as the final feature representation, resulting in redundancy and poor feature representation. Although ViT overcomes the drawback of CNNs in capturing long-distance dependencies, it still exhibits weaknesses in inducing bias (*Jia et al., 2022*).

To overcome the aforementioned challenges and facilitate the simultaneous processing of image and voice features, we present a scoring model for oral English teaching based on multi-modal perception using the Internet of Things, leveraging the ViT framework. Initially, a CNN model is trained to extract image information and audio features from the Internet of Things, which are subsequently quantified to a fixed dimension. Subsequently, we introduce a transformer-based approach to attend to both spoken English features and image features. Finally, the English teaching content is classified and graded based on the quantitative characteristics of oral dialogue. Our proposed method outperforms state-of-the-art (SOTA) techniques, demonstrating excellent scoring performance for oral English teaching. The key contributions of our study are as follows:

We address the challenge of processing images and voices for supervision in oral English teaching by proposing a scoring model based on multi-modal perception using the Internet of Things.

We introduce the External Attention and Feature selection module, which effectively attends to the fused features and enhances accuracy, resulting in SOTA performance.

## RELATED WORK

Vision Transformer adopts a multi-layer Transformer architecture to complete the feature extraction process, which uses self-attention as the feature function (*He et al., 2021*). Then, it exploits the posterior layer Transformer to refine the previous layer's output.

Firstly, images should be sliced into a series of patches. $x_p \in \mathbb{R}^{N \times (P \times P \times C)}$ By the non-overlapping way. Note that $H \times W$ is the size of resolutions in an image, $C$ represents the number of channels, $P \times P$ is regarded as the resolution, and $N = HW/P^2$, is a number of patches. Subsequently, each of the patches is mapped to the size of D-d space by learnable linearization mapping vector $E \in \mathbb{R}^{(P \times P \times C) \times D}$. Subsequently, the classification vector $x_{class} \in \mathbb{R}^{1 \times D}$ is added to the patch sequence header to integrate global features. Finally, the input of the first-layer Transformer $Z_0$ is obtained by assigning position features to each patch with $E_{pos} \in \mathbb{R}^{(P \times P \times C) \times D}$.

$$Z_0 = \left[ x_{class}, x_p^1 E, x_p^2 E, \ldots, x_p^N E \right] + E_{pos} \tag{1}$$

After inputting $Z_0$ to the first Transformer layer, the Multi-head Attention (MHSA) and Multi-layer Perceptron (MLP) modules with the residual structure are respectively adopted for feature extraction. Features are normalized by Layernorm (LN) before input to these two modules.

To extract refining and sufficient information, ViT uses a multi-layer Transformer architecture to refine the output features of the previous layers, the formula is:

$$\tilde{Z}_t = \text{MSA}(\text{LN}(Z_{t-1})) + Z_{t-1}, t = 1, 2, \ldots, T \tag{2}$$
$$Z_t = \text{MSA}\left(\text{LN}(\tilde{Z}_t)\right) + \tilde{Z}_t, t = 1, 2, \ldots, T \tag{3}$$

where $t$ is the number of layers, $\tilde{Z}_t$ and $Z_t$ are the results of the features processed by the MSA and MLP modules in the $t$-th layer, respectively. The global characteristics of the input image are gradually refined and aggregated into the class patch by the flow process in the multi-layer Transformer. Therefore, the class patches $Z_T^0$ of last layer output are processed by Layernorm: $y = \text{LN}(Z_T^0)$ Which are the final global features $y$. The classification prediction, loss calculation, and backpropagation are input into the classifier to complete the model's training.

## A SCORING MODEL OF ORAL ENGLISH TEACHING BASED ON MULTI-MODAL PERCEPTION

In this article, we employ the Internet of Things (IoT) as a means to utilize audio, video sensors, and other devices for real-time collection of sounds and images from monitoring or interactive objects. By incorporating IoT, we gather images and voices as the foundation of our research. To address the challenge of extracting segmentation edge features, we introduce a modification to the Vision Transformer (ViT) model by replacing the non-overlapping method with a sliding window approach. This enables us to generate sequences of image patches, thereby mitigating the difficulties associated with extracting precise edge features.
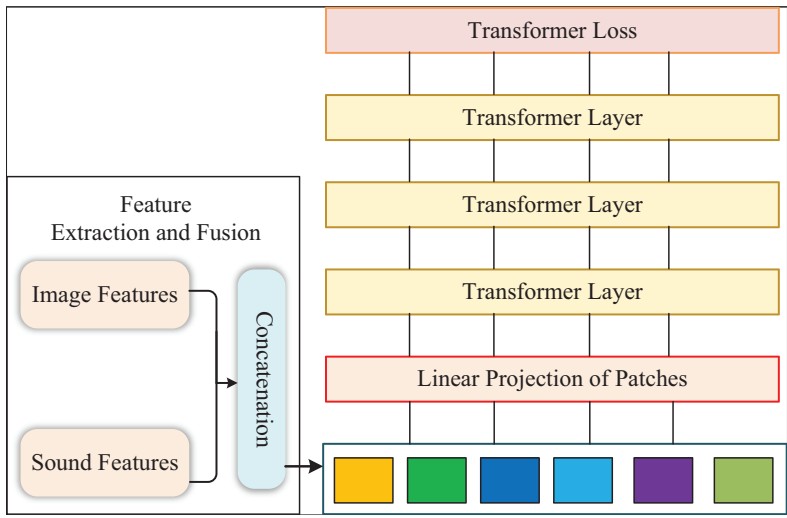

**Figure 1  The structure of our method.**     

Regard the image resolution as $H \times W$ and the calculation method of patch number $N$; the formula can be represented as:

$$N = \frac{H - P + S}{S} \times \frac{W - P + S}{S}. \tag{4}$$

The framework of our model is presented in Fig. 1. Firstly, we add $x_{dis} \in \mathbb{R}^{1 \times D}$ to calculate the multivariate loss at the end of the patch sequence. Then, we propose the external attention to dropout the self-attention for each Transformer layer. Subsequently, we adopt the feature selection module to filter the input of the last Transformer layer, which can remove redundant features. Finally, we apply the output of the last-layer Transformer to calculate the loss from various aspects and fuse them.

### External attention

Figure 2 illustrates the intricate architecture of the self-attention module, which serves as the fundamental feature processing method in ViT. It is important to note that we opted for the Transformer model instead of convolutional neural networks (CNN) or long short-term memory (LSTM) because we need to process distinct types of features, namely images and voices. CNN or LSTM alone would not suffice for this purpose. Furthermore, the Bert model is well-suited for handling large datasets. Firstly, we linear the fused features $F \in \mathbb{R}^{\tilde{N} \times d}$ ($\tilde{N}$ represents the account of pixels, and $d$ refers to the dimension of the fused features) to $Query \in \mathbb{R}^{\tilde{N} \times d}$, $Key \in \mathbb{R}^{\tilde{N} \times d}$ and $Value \in \mathbb{R}^{\tilde{N} \times d}$. Then, we calculate the attention weighting matrix by $Query$ and $Key$. The specific calculation process is as follows:

$$\alpha_{i,j} = \text{softmax}\left(Query \otimes Key^{\mathrm{T}}\right) \tag{5}$$

where $\alpha_{i,j}$ Calculates a similarity score between $i$-th pixel and $j$-th pixel. In addition, we multiply. $\alpha_{i,j}$ By $Value$ and make a residual connection for $F$ to obtain the final output feature. The formula is as follows:

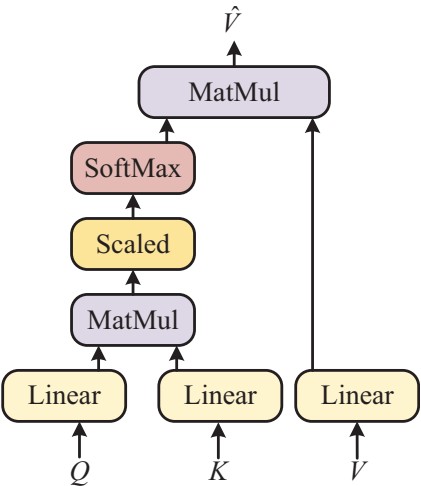

**Figure 2 Self-attention model.**

$$F_{out} = F + \alpha_{i,j} \otimes Value, F_{out} \in \mathbb{R}^{\tilde{N} \times d}. \tag{6}$$

In the aforementioned calculation process, the inherent correlation between samples is overlooked, leading to insufficient feature extraction for the model. Most pixels within an individual sample exhibit correlations with only a few other pixels. Enumerating these correlations results in redundant calculations and an excessive number of model parameters.

To address the limitations of self-attention, we introduce an external attention mechanism. This mechanism enhances the model by capturing correlations among intra-sample and inter-sample elements through two learnable external memory units. By incorporating external attention, we improve the model's feature extraction capabilities while reducing the number of parameters required for computation.

The detailed structure is shown in Fig. 3. Firstly, we map the feature map $F \in \mathbb{R}^{\tilde{N} \times d}$ to the vectors $Q_E \in \mathbb{R}^{\tilde{N} \times d}$. Then, we use the product of a learnable external memory unit. $M_k \in \mathbb{R}^{S \times d}$ and $Q_E$ to obtain the attention weighting map $A_E$ By the regularization. The formula is as follows:

$$A_E = Norm(Q_E \otimes M_k). \tag{7}$$

Subsequently, we apply the $A_E$ and another external memory component $M_v \in \mathbb{R}^{S \times d}$ To compute another more refined feature map. The final output $F_{out}$ It is obtained by performing residual operations on the input features. The formula is as follows:

$$F_{out} = F + A_E \otimes M_v. \tag{8}$$

## Feature selection

In the sliding window method, it is possible for certain patches to contain only background information or capture a small portion of foreground objects. Decreasing the step size of

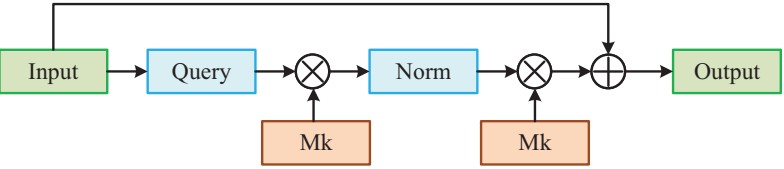

**Figure 3 External-attention model.**

the sliding window increases the occurrence of similar patches. Given the abundance of discriminative features with subtle variations in images and voices, including insufficient information from multiple patches can lead to redundant output features.

To tackle this issue, we propose the integration of a feature selection module. This module selectively removes discriminatory parts of images and voices to alleviate redundancy in the output. By incorporating the feature selection module, we can enhance the effectiveness of feature extraction by focusing on the most informative and discriminative aspects of the data. We cannot select the discriminative patches without attention to weight information for the original patch sequence. According to the attention weight of the previous $L - 1$ layers, the feature selection module can select the discriminative features output by the $(L - 1)$-th layer. Besides, we use them as the layer input to further refine the features. Finally, we regard the $(L - 1)$-th layer's output of the Transformer as shown in the formula:

$$Z_{L-1} = \left[ Z_{L-1}^0, Z_{L-1}^1, \ldots, Z_{L-1}^N, Z_{L-1}^{N+1} \right]. \tag{9}$$

The attending weights of the previous $L - 1$ layer are regarded as follows:

$$\alpha_l = \left[ \alpha_l^0, \alpha_l^1, \ldots, \alpha_l^K \right], l = 1, 2, \ldots, L - 1. \tag{10}$$

$$\alpha_l^i = \left[ \alpha_l^{i0}, \alpha_l^{i1}, \ldots, \alpha_l^{iN}, \alpha_l^{iN+1} \right], i \in [0, K]. \tag{11}$$

where $K$ refers to the accounts of attention heads, and then, we feed the continuous multiplication operation into the feature selection module to integrate the attended weightings for the previous $L - 1$ layers, which can be shown in the formula:

$$\alpha = \prod_{l=0}^{L-1} \alpha_i. \tag{12}$$

where $\alpha$ records the transfer process of attention weight from the first layer to the $(L - 1)$-th layer, we choose the $k$ indexes. $[A_1, A_2, \ldots, A_K]$. They correspond to the maximum value of each attention head. Finally, the features are chosen to input them to the $L$-th layer according to the index. The selected sequence can be expressed as follows:

$$Z = \left[ Z_{L-1}^0, Z_{L-1}^{A_1}, Z_{L-1}^{A_2}, \ldots, Z_{L-1}^N, Z_{L-1}^{N+1} \right]. \tag{13}$$

where $Z$, regarded as the input of the final layer, drops too many invalid features extracted to avoid the redundancy of the last output features.
## Fused multivariate loss function

To address the subtle differences between different classes of images and voices, as well as the significant differences within the same class, we propose a fused multivariate loss function that optimizes from multiple perspectives. This approach aims to overcome the weak bias induction ability of Transformer models.

In our proposed approach, we utilize a combination of cross-entropy loss and contrastive loss. The cross-entropy loss is employed to capture the noticeable inter-class differences. On the other hand, the contrastive loss serves to increase the dissimilarity between features of subclasses while reducing the disparity among features of the same subclass. In this article, we maintain the calculation of the cross-entropy loss based on the class patch output from the last Transformer. Simultaneously, we calculate the contrastive loss using the same patch. The calculation process can be expressed as follows:

$$\tau_1 = \frac{1}{N_B^2} \sum_i^{N_B} \left[ \sum_{j:y_i=y_i}^{N_B} \left(1 - \cos(Z_i, Z_j)\right) + \sum_{j:y_i \neq y_i}^{N_B} max\left(\left(\cos(Z_i, Z_j) - \alpha\right), 0\right) \right] \tag{14}$$

where $N_B$ Is the size of the batch, $Z_i$ It is the class patch output of the $i$-th image through the model, which is also the final feature representation. Besides, $\cos(Z_i, Z_j)$ represents the cosine similarity of $Z_i$ and $Z_j$, which will play a role in the contrast loss when it is greater than the hyperparameter $\alpha$. Through backpropagation, $\tau_1$ can extend the feature representation between different subcategories and reduce the feature representation within the same subcategory, alleviating the classification difficulty caused by small between-class and large within-class differences.

In addition to the previously mentioned loss functions, we recognize the importance of inductive bias in affecting feature extraction within the Transformer model. Given that convolutional neural networks (CNNs) possess a strong inductive bias, we introduce a distillation loss from the CNN to enhance the feature extraction process. In this article, we incorporate the distillation loss as part of the total loss function. We introduce a distillation patch, which is added after the input patch sequence. Similar to the class patch, the distillation patch interacts with other patches in multiple Transformer layers, and the features of the image are subsequently aggregated. However, unlike the class patch, the distillation patch reproduces the predicted labels generated by the Teacher model (CNN), rather than the actual labels.

To achieve the distillation loss, we calculate the Kullback–Leibler (KL) divergence based on the combination of the labels calculated by the distillation patch and the output labels of the Teacher Model (CNN). By incorporating the distillation loss, we enable the Transformer model to learn the inductive bias from the CNN, thereby significantly improving the feature extraction process. As the partial module of our loss function, the Student model is guided to perform backpropagation by the loss above. The specific calculation method is as follows:

$$\tau_2 = \varphi^2 KL\left( \text{softmax}\left(\frac{Z_{stu}}{\varphi}\right), \text{softmax}\left(\frac{Z_{tea}}{\varphi}\right) \right) \tag{15}$$

where $Z_{stu}$ Is the output of the Logist function when using a distillation patch for classification, $Z_{tea}$, is the output of the logits function of the Teacher model, and $\varphi$ represents the distillation temperature.

To summarize, we incorporate three loss functions—cross-entropy loss, contrast loss, and distillation loss—from different perspectives. These losses are fused together to enhance the model's ability to distinguish between subclasses, reduce differences within the same subclass, and provide the model with inductive bias. This fusion process results in more refined and discriminative output features.

The cross-entropy loss and contrast loss are calculated based on the same class patch, and thus, they are considered as a part of the total loss in the fusion process. On the other hand, the distillation loss is treated as a separate component. The fusion method we employ can be described as follows:

$$\tau = (1 - \gamma)(\tau_{CE}(y', y) + \tau_1(Z)) + \gamma\tau_2 \tag{16}$$

where $\tau_{CE}(y', y)$ Is the cross-entropy loss for the predicted $y'$ As well as the true $y$, according to the class patch, and $\gamma$ is the hyperparameter.

# EXPERIMENT AND ANALYSIS

## Datasets

The dataset of oral English teaching scoring includes 50,000 images and corresponding voices, which are from the videos of cram school on the IoT. We choose 40,000 pairs of image and voice as the training data, 5,000 pairs as validation, and 5,000 as test sets. In addition, we apply horizontal flip, vertical flip, and auto-augment to expand the dataset, which can avoid overfitting the model due to too little data.

## Implement details

We implemented our experiments with the i7-12900k Cpu, 3080ti Gpu and the Pytorch deep learning framework. The training settings of the scoring model for oral English teaching based on the multi-modal perception of the Internet of Things are represented in Table 1.

We choose accuracy as the evaluation metric, which can be presented as:

$$Accuracy = \frac{I_c}{I_{total}} \tag{17}$$

where $I_c$ refers to the accounts of correct classification images and $I_{total}$, refers to the counts for the total images.

In this article, we utilize a 12-layer structure for the Transformer model. The number of patches has a significant impact on the model's parameter count, inversely proportional to the patch size and proportional to the input image's resolution. To ensure sufficient data for the model evaluation and training convergence, we use a resolution of $448 \times 448$ for input images. During the training phase, we employ random cropping, while central cropping is used during the testing phase. The patch size of $16 \times 16$ from the original ViT model is retained, and the sliding window step is set to 12. In the training phase, we set the

**Table 1** Training settings.

| Hyper parameters | The values |
|---|---|
| Learning rate | 0.0004 |
| Dropout | 0.1 |
| Model optimizer | Adam |
| Transformer size | 12 |
| Number of epoch | 250 |
| Batch size | 30 |

hyperparameter α to 0.4. For optimization, we employ stochastic gradient descent (SGD) with a momentum setting of 0.9. The batch size is set to 32, and the initial learning rate for training is 0.03. We also incorporate cosine annealing to control the learning rate's decline.

To gather information, we collect images, audio, and video from various sensors in the IoT. We then perform feature extraction using ResNet, ensuring that the feature extraction model satisfies the aforementioned training parameters. Subsequently, we concatenate the extracted image features and voice features before inputting them into the Transformer model. Finally, these features are fed into the Transformer to obtain the desired results.

## Experiments results

### Ranks of classification division

Before performing the experimental performance comparison, we must artificially set some scoring ranks and unify the classification criteria. We also assign accurate classes to the features in the dataset. To realize it, we artificially divide the scoring positions into excellent, good, medium, pass, and poor in Table 2.

Furthermore, to establish the ground truth and evaluate the performance of our model, we manually scored and classified 50,000 data groups. This process allowed us to obtain the score distribution and statistics for the entire dataset, as depicted in Figs. 4 and 5.

From the analysis of Figs. 4 and 5, we observed that the majority of the data in the dataset fell into the categories of "Excellent" and "Medium," while the classes of "Ideal," "Pass," and "Poor" accounted for a smaller portion. Based on this observation, we determined that using five ranks for classification would be suitable. The distribution of ranks shown in Fig. 4 was deemed optimal. It is important to note that the number of samples in the "Excellent" and "Poor" ranks is relatively low, making further subdivisions impractical.

### Comparison results of different models

To further verify our proposed method, we choose some classical methods to conduct comparative experiments on oral English teaching scoring tasks, such as LSTM, DB, DVAN, RA-CNN, MC Loss, and ViT, as shown in Table 3.

Table 3 showcases the impressive performance of our proposed Oral English Teaching scoring method on the Oral English Teaching scoring dataset. With a remarkable accuracy rate of 88.8%, our approach outperforms other models and achieves a new state-of-the-art

| Table 2 Ranks of scoring. | |
|---|---|
| **Ranks** | **Values** |
| Excellent | 90~100 |
| Good | 75~89 |
| Medium | 60~75 |
| Pass | 40~59 |
| Poor | <40 |

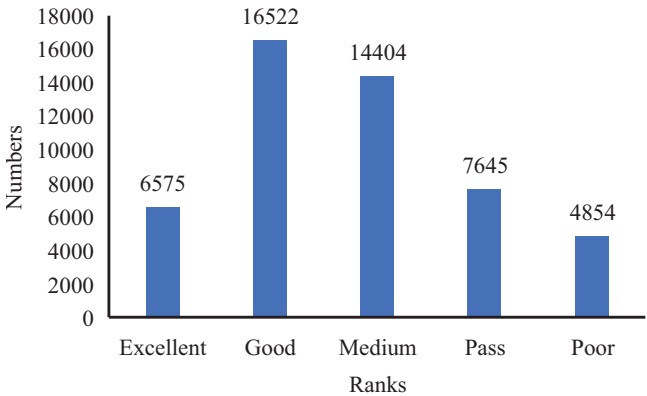

**Figure 4** Ranks distribution in oral English teaching scoring dataset.

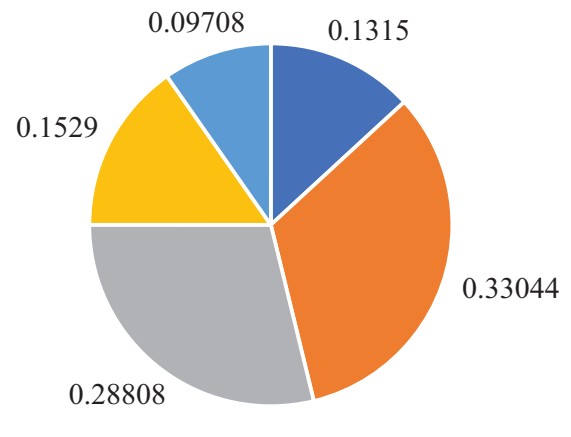

**Figure 5** Ranks statistics in oral English teaching scoring dataset.

performance. When compared to LSTM, our method exhibits a significant improvement of 14.7% in accuracy. This notable advancement can be attributed to the superior capabilities of the Transformer architecture in handling multi-modal features compared to the RNN architecture. In addition, our approach surpasses two CNN-based methods, DB and RA-CNN, by more than 3% in terms of accuracy, highlighting its effectiveness in

**Table 3 Comparison with other methods based on deep learning framework.**

| Methods | Base model | Accuracy (%) |
|---|---|---|
| LSTM | RNN | 74.1 |
| DB (*Sun et al., 2020*) | ResNet-50 | 85.6 |
| DVAN (*Zhao et al., 2017*) | VGG-16 | 77.0 |
| RA-CNN (*Fu, Zheng & Mei, 2017*) | VGG-19 | 82.5 |
| MC Loss (*Chang et al., 2020*) | ResNet-50 | 84.3 |
| ViT | SA | 86.8 |
| Ours | EA | 88.8 |

extracting and leveraging feature information. Furthermore, our method outperforms the DVAN model by 10% in accuracy, further emphasizing its superiority and capability in delivering higher accuracy levels compared to existing models. Finally, when compared to ViT, our proposed method demonstrates an improvement of over 2% in accuracy. This improvement is primarily attributed to the incorporation of external attention enhancements in our approach.

In conclusion, our proposed method not only achieves exceptional accuracy in oral English teaching scoring but also surpasses other existing models in performance. Its outstanding accuracy, coupled with its superiority over other methods, establishes its effectiveness and suitability for accurate and reliable oral English teaching scoring applications.

### Ablation experiment

We carried out the ablation experiments to evaluate the advancement of each module on the Oral English teaching scoring dataset by gradually integrating each module into ViT.

Table 4 displays the results of our ablation experiments, where we compare the performance of different Transformer network structures using only the cross-entropy loss function. The following four configurations are evaluated:

ViT (baseline): This represents the original Transformer model, as described in the ViT paper.

ViT (EA): In this configuration, we substitute the self-attention module with the external attention module proposed in our method.

ViT (FS): This configuration incorporates the feature selection module into the ViT model.

ViT (EA&FS): Here, we combine both the external attention module and the feature selection module in the ViT model.

The experimental results demonstrate the effectiveness of each modification, and the corresponding performance metrics are presented in Table 4.

As shown in Table 4, the results highlight the impact of our proposed modifications using the cross-entropy loss function. The external attention (EA) module improves the accuracy of ViT by 0.8%. The feature selection (FS) module further enhances the accuracy to 87.8% compared to ViT. When both the EA module and FS module are applied, our

**Table 4 Ablation experiment results.**

| Methods | Composition | Accuracy (%) |
|---|---|---|
| ViT (Baseline) | SA | 86.8 |
| ViT (EA) | EA | 87.6 |
| ViT (FS) | SA + FS | 87.8 |
| ViT (EA + FS) | EA + FS | 88.8 |

method achieves a significant improvement of 2% over ViT, reaching an accuracy of 88.8%. By leveraging audio and video data from IoT, we have enhanced the attention mechanism and fusion loss function based on CNN and ViT. This has led to the identification of key factors in English curriculum and accurate evaluation of English teaching.

## CONCLUSION

The proposed scoring model for oral English teaching based on the multi-modal perception of the Internet of Things addresses the need for effective scoring and supervision in online oral English teaching. By leveraging IoT technologies, we extract audio and video features using a pre-trained CNN model. We then employ Transformer-based methods for processing spoken English features and image features separately, highlighting the importance of spoken English features. Based on quantitative elements of oral dialogue, we classify and grade English teaching content. Comparisons with other classification methods demonstrate the superior performance of our scoring model, achieving the highest accuracy rate of 88.8% and surpassing other methods by over 2%. This signifies the commercial value and practical applicability of our approach in scoring oral English teaching in IoT environments. Future research directions include expanding the scope of oral English teaching supervision to encompass all teaching domains. Additionally, we aim to further enhance our attention mechanisms to identify and attend to more crucial aspects of oral English teaching, ultimately improving prediction accuracy.

### Funding
The authors received no funding for this work.

### Competing Interests
The authors declare that they have no competing interests.

### Author Contributions
- Haitao Qin conceived and designed the experiments, performed the experiments, analyzed the data, performed the computation work, prepared figures and/or tables, authored or reviewed drafts of the article, and approved the final draft.

## Data Availability

The code is available in the Supplemental File.

The data is available at Zenodo: Elisabeth Mayweg-Paus, Maria Zimmermann, & Theresa Ruwe. (2021). DIALLS Dataset of evaluations of Open Educational Resources—The Cultural Literacy Learning Programme resources [Data set]. Zenodo. https://doi.org/10.5281/zenodo.5553923.

## Supplemental Information

Supplemental information for this article can be found online at http://dx.doi.org/10.7717/peerj-cs.1503#supplemental-information.

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
