# Peer review of "Design of oral English teaching model based on multi-modal perception of the Internet of Things and improved conventional neural networks"

_PeerJ Computer Science, doi:10.7717/peerj-cs.1503_

## Round 0.1 · original submission · Major Revisions

Dear Author,

Please see the reviewers' detailed comments. Reviewer 1 provided valuable feedback on the article. They highlighted the need for a clear explanation of the Internet of Things (IoT) and its role in the methodology, as well as a more specific problem statement and emphasis on the significant contribution of the research in the introduction. The reviewer suggested including key formulas to clarify references to discriminant analysis, logistic regression, and other methodologies in the related work section. They also requested an explanation for the authors' preference for the Transformer model over other alternatives like Bert, CNN, or LSTM for the classification task.

Reviewer 2 discussed several issues that need to be addressed. These include the need for English language revisions, highlighting the contribution of the oral English teaching scoring methods, addressing the study's originality and contribution to the literature, explaining the choice of five ranks for scoring, providing explanations for symbols and abbreviations, analyzing the results of ablation experiments, detailing the processing of image and voice information in the Transformer model, revising the conclusion to focus on the paper's contribution, and adding a research plan for future oral English teaching scoring.

Reviewer 1 ·

Basic reporting

In the post-pandemic era, the teaching of spoken English encompasses a distinct mode and methodology, necessitating urgent research on an intelligent scoring approach that can supervise the content of English instruction. This paper presents a scoring methodology for oral English instruction, founded upon the multi-modal perception capabilities of the Internet of Things. The proposed method optimizes the network model to cater to the task of scoring oral English instruction, thereby enhancing the model's robustness and augmenting the accuracy of the oral English scoring classification.

Experimental design

Check comments section

Validity of the findings

Check comments section

Additional comments

This innovative approach effectively resolves the temporal fusion challenge associated with integrating image and voice information within the Internet of Things domain. Furthermore, it facilitates timely feedback for teaching professionals, thereby bestowing significant practical value. Once these aforementioned aspects are addressed, I recommend the acceptance of this article.

1. Please elucidate the essence of the Internet of Things (IoT) and expound upon its pivotal role in the methodology.
2. What distinguishes this paper as a remarkable contribution? The introduction lacks clarity regarding the specific problem addressed in this article. Furthermore, it is imperative to introduce the significant contribution made by this research.
3. In the section concerning related work, it would be beneficial for the authors to enumerate key formulas to elucidate the references pertaining to discriminant analysis, logistic regression, and other relevant methodologies.
4. Kindly provide an explanation for the authors' preference for the Transformer model over alternatives such as the Bert model, CNN, or LSTM, which may seemingly be more suitable for the classification task at hand.
5. Could you please delineate the experimental conditions, encompassing the equipment, operating system, and deep learning framework employed in the design of the proposed methods?
6. The aforementioned considerations should be included in the Implementation Details section to facilitate the replication of the research by other scholars.
7. The Conclusion section necessitates enhancement. The authors ought to provide a comprehensive summary of their entire body of work, elucidating the achieved results, advantages, disadvantages, and also propose a few potential future directions.

The authors should be attentive to articulating their own innovative accomplishments and avoid blurring the distinction between the achievements of others and their own contributions, thereby eliminating any ambiguity or contradictions pertaining to the innovative aspects.

Reviewer 2 ·

Basic reporting

Oral English teaching is an important part of education and teaching tasks. In this paper, a scoring model for oral English teaching based on multi-modal perception of the Internet of Things is proposed to improve the accuracy of prediction and understand the state of the oral English teaching. In practical application, the method in this paper can be applied with scoring of oral English teaching on the Internet of things and have great commercial value., which is helpful to predict the status of oral English teaching and make supervision in advance.
However, there are also some problems, some revisions needed to be revised to make sure that the manuscript can be accepted. The commonly problems are as follows:

1. The English of your manuscript should be revised in any section of the paper before resubmission. We strongly suggest that you obtain assistance from a colleague who is well-versed in English or whose native language is English.

2. I suggest that it is necessary to add the contribution of oral English teaching scoring methods in the first section, which can help readers to understand this paper.

3. The originality of the study and its contribution to the literature have not been adequately addressed.

4. As we known, the number of ranks can determine the performance of the oral English teaching scoring. Please introduce why the number of ranks is 5, not the 6 or more?

5. Each symbol/abbreviation must be explained, when it is used for the first time in the paper text.

6. We can see that the ablation experiments are conducted. However, the authors have not analyzed the results, which can help us understand the performance of modules.

7. How do the authors process image information and voice information for the Transformer? Please introduce the strategy of feature inputting model, such as adding, concatenation or multiplication.

8. The conclusion section should be revised carefully to concentrate on the contribution of this paper.

9. Please check it and add the research plan for Oral English teaching scoring in the future.

Experimental design

See above

Validity of the findings

See above

Additional comments

See above

---

## Round 0.2 · accepted · Accept

I confirm that the authors have addressed all of the reviewers' comments.

Reviewer 1 ·

Basic reporting

The manuscript is updated and all my questions are answered well and incorporated the suggested changes. I have no further concerns and accept the manuscript for publication.

Experimental design

Improved

Validity of the findings

Improved

Reviewer 2 ·

Basic reporting

The authors have revised the paper according to my previos comments, therefore, i have no more comments.

Experimental design

The authors have revised the paper according to my previos comments, therefore, i have no more comments.

Validity of the findings

the findings of the research are valid and enhances the body of the knowledge.

Additional comments

No more additional comments.